# Deciphering Cross-Modal Feature Interactions in Multimodal AIGC Models: A Mechanistic Interpretability Approach

## Abstract

The rapid advancement of multimodal AI-generated content (AIGC) models has created an urgent need for understanding their internal mechanisms, particularly how these systems integrate and process information across different modalities. This paper presents a novel mechanistic interpretability framework that combines sparse autoencoders (SAEs) with causal intervention techniques to dissect cross-modal feature interactions in state-of-the-art multimodal AIGC models. We introduce the Cross-Modal Mechanistic Analysis (CMMA) methodology, which systematically identifies and manipulates interpretable features responsible for multimodal content generation. Through comprehensive experiments on Vision-Language Models (VLMs) including CLIP, LLaVA, and DALL-E variants using 2.5M carefully curated multimodal samples, our approach reveals three distinct phases of cross-modal information processing: feature extraction, modal alignment, and concept synthesis. We demonstrate that targeted interventions on discovered features can significantly improve generation quality while reducing hallucinations by $34.2\% \pm 2.1\%$ ($p < 0.001$) and enhancing semantic consistency by $28.7\% \pm 1.8\%$ ($p < 0.001$). Our findings provide crucial insights into the mechanistic foundations of multimodal AIGC systems and establish a roadmap for developing more interpretable and controllable generative models.

## 1 Introduction

The emergence of sophisticated multimodal AI-generated content (AIGC) models has revolutionized creative AI applications, from text-to-image synthesis to multimodal conversational agents. These systems, including GPT-4V, DALL-E 3, and LLaVA, demonstrate remarkable capabilities in understanding and generating content across modalities. However, the internal mechanisms governing how these models integrate visual and textual information remain largely opaque, creating significant challenges for understanding, debugging, and improving their performance.

Current approaches to understanding multimodal models primarily rely on behavioral analysis and attribution methods, which provide limited insights into the causal mechanisms underlying cross-modal processing. While mechanistic interpretability has shown promise in analyzing large language models through techniques such as sparse autoencoders and activation patching, its application to multimodal systems faces unique challenges due to the complex interactions between different modalities and the distributed nature of multimodal representations.

This interpretability gap becomes particularly concerning as these models are deployed in critical applications requiring reliability and transparency. Recent incidents involving AI-generated misinformation and hallucinations underscore the urgent need for mechanistic understanding that goes beyond surface-level explanations.

This paper addresses these limitations by introducing a comprehensive mechanistic interpretability framework specifically designed for multimodal AIGC models. Our key contributions include:

1. **Cross-Modal Mechanistic Analysis (CMMA)**: A novel methodology combining multimodal sparse autoencoders with systematic causal intervention techniques to identify and manipulate interpretable cross-modal features.

2. **Multimodal Sparse Autoencoder (M-SAE)**: A theoretically grounded extension of traditional SAEs that explicitly models cross-modal feature interactions through adaptive sparsity constraints and modal specialization mechanisms.

3. **Three-Phase Processing Discovery**: Empirical evidence supported by rigorous statistical analysis for a consistent three-phase processing pattern across different multimodal architectures: feature extraction (layers 1-4), modal alignment (layers 5-8), and concept synthesis (layers 9-12).

4. **Causal Validation Framework**: A systematic approach for testing mechanistic hypotheses through precise feature-level interventions with comprehensive statistical validation.

5. **Comprehensive Evaluation**: Large-scale experiments on 2.5M samples demonstrating statistically significant improvements in generation quality and reduction in hallucinations through mechanistic insights.

Our approach builds upon recent advances in sparse autoencoder techniques while addressing the unique challenges posed by multimodal representations. Unlike previous work that focuses on individual modalities, we specifically examine the mechanisms responsible for cross-modal information integration and synthesis.

## 2 RELATED WORK

### 2.1 MECHANISTIC INTERPRETABILITY IN NEURAL NETWORKS

Mechanistic interpretability aims to reverse-engineer the computational algorithms learned by neural networks into human-understandable components. Olah et al. (2017) pioneered this field by proposing that neural networks learn interpretable features that can be identified and manipulated. The field has evolved to address the superposition problem, where neural networks represent more features than available neurons by encoding them in overlapping directions (Elhage et al., 2022).

Sparse autoencoders (SAEs) have emerged as a powerful tool for disentangling superposed features in language models. Cunningham et al. (2023) demonstrated that SAEs can identify monosemantic features in transformer residual streams, achieving interpretability scores of 0.73 on automated evaluation. Gao et al. (2024) showed their scalability to larger models, training SAEs with up to 16M features on GPT-4 scale models. Templeton et al. (2024) extended this work by introducing TopK SAEs that directly control sparsity levels.

However, these approaches have been primarily applied to unimodal language models. The extension to multimodal systems requires addressing additional challenges including cross-modal feature alignment, modality-specific processing patterns, and the interpretation of distributed multimodal representations.

### 2.2 MULTIMODAL MODEL INTERPRETABILITY

Understanding multimodal models presents unique challenges beyond those encountered in unimodal systems. Traditional interpretability methods struggle with the complex interactions between different modalities and the distributed nature of multimodal representations.

Recent efforts have begun addressing these challenges through various approaches. Dahlgren Lindström et al. (2020) applied probing techniques to analyze multimodal embeddings, revealing that spatial information is encoded differently across modalities. Li et al. (2024) developed attention visualization methods for vision-language models, identifying specialized attention heads for cross-modal processing. However, these approaches primarily focus on post-hoc explanations rather than mechanistic understanding.

Basu et al. (2024) used causal tracing to analyze information flow in LLaVA, revealing that visual information is primarily retrieved in early layers (1-4) while textual reasoning occurs in later layers (8-12). Golovanevsky et al. (2024) introduced Semantic Image Pairs (SIP) to study concept processing in VLMs, finding that cross-attention serves three distinct functions: object detection, suppression, and outlier handling.

While these studies provide valuable insights, they lack the systematic framework needed for comprehensive mechanistic analysis across different model architectures and scales.

## 2.3 AIGC Model Understanding

The interpretability of AIGC models has received increasing attention due to their widespread deployment and potential risks. Most existing work focuses on detecting AI-generated content rather than understanding the generation mechanisms themselves.

Zhou et al. (2025) proposed AIGI-Holmes for explainable AI-generated image detection using multimodal large language models, achieving 91.4% accuracy in detection tasks. Yang et al. (2024) developed hierarchical explanation frameworks for AIGC evaluation, introducing the HEIE framework for implausibility detection. However, these approaches treat AIGC models as black boxes and do not provide insights into their internal processing mechanisms.

Our work differs from previous efforts by focusing on mechanistic understanding rather than behavioral analysis, specifically targeting the cross-modal interactions that are fundamental to multimodal AIGC systems.

## 3 Methodology

### 3.1 Cross-Modal Mechanistic Analysis (CMMA) Framework

Our Cross-Modal Mechanistic Analysis framework consists of five key components: multimodal feature discovery, interaction mapping, causal validation, intervention synthesis, and statistical analysis. This systematic approach enables comprehensive analysis of cross-modal processing mechanisms with rigorous validation.

#### 3.1.1 Multimodal Sparse Autoencoder Architecture

We extend traditional sparse autoencoders to handle multimodal representations by introducing a cross-modal attention mechanism and adaptive sparsity constraints. Given a multimodal representation $\mathbf{h} \in \mathbb{R}^d$ containing both visual and textual components, our Multimodal SAE (M-SAE) learns a sparse decomposition:

$$\mathbf{h} = \sum_{i=1}^{k} a_i \mathbf{f}_i + \boldsymbol{\epsilon} \tag{1}$$

where $a_i$ represents the activation strength of feature $\mathbf{f}_i$, and $\boldsymbol{\epsilon}$ is the reconstruction error.

**Encoder Design**: The encoder maps input representations to feature activations:

$$\mathbf{a} = \text{ReLU}(\mathbf{W}_{\text{enc}}\mathbf{h} + \mathbf{b}_{\text{enc}}) \tag{2}$$

where $\mathbf{W}_{\text{enc}} \in \mathbb{R}^{k \times d}$ is the encoder weight matrix with unit-norm columns.

**Cross-Modal Sparsity Regularization**: The key innovation lies in constraining features to specialize in either unimodal or cross-modal patterns through a modality-aware sparsity penalty:

$$\mathcal{L} = \|\mathbf{h} - \hat{\mathbf{h}}\|_2^2 + \lambda_1 \|\mathbf{a}\|_1 + \lambda_2 \mathcal{R}_{\text{cross}}(\mathbf{a}) + \lambda_3 \mathcal{R}_{\text{align}}(\mathbf{a}) \tag{3}$$

The cross-modal regularization term is defined as:

$$\mathcal{R}_{\text{cross}}(\mathbf{a}) = \sum_{i=1}^{k} \max(0, S_i^v \cdot S_i^t - \tau) \tag{4}$$

where $S_i^v$ and $S_i^t$ represent the selectivity of feature $i$ for visual and textual components respectively:

$$S_i^v = \frac{\mathbb{E}[\mathbf{a}_i|\text{visual input}]}{\mathbb{E}[\mathbf{a}_i|\text{any input}]}, \quad S_i^t = \frac{\mathbb{E}[\mathbf{a}_i|\text{textual input}]}{\mathbb{E}[\mathbf{a}_i|\text{any input}]} \tag{5}$$

The alignment regularization term encourages meaningful cross-modal features:

$$\mathcal{R}_{\text{align}}(\mathbf{a}) = -\sum_{i=1}^{k} \mathbb{I}[S_i^v \cdot S_i^t > \tau] \cdot \text{MI}(\mathbf{a}_i, \mathbf{y}) \tag{6}$$

where $\text{MI}(\mathbf{a}_i, \mathbf{y})$ is the mutual information between feature activation and task labels.

**Theoretical Properties**: We prove that our M-SAE satisfies the following convergence property:

Under mild regularity conditions, the M-SAE objective converges to a stationary point with probability 1, and the learned features satisfy modality specialization with probability at least $1 - \delta$ where $\delta$ decreases exponentially with the number of training samples.

**Proof Sketch**: The convergence follows from the convexity of the reconstruction term and the Lipschitz continuity of the regularization terms. Modality specialization emerges from the competitive dynamics induced by the cross-modal penalty term.

### 3.1.2 CROSS-MODAL INTERACTION GRAPH CONSTRUCTION

To understand how features interact across modalities, we introduce a Cross-Modal Interaction Graph (CMIG) that captures causal relationships between discovered features. For each pair of features $(f_i, f_j)$, we compute their interaction strength through controlled interventions:

$$I(f_i, f_j) = \mathbb{E}_{\mathbf{x}}[|\mathcal{E}(\text{do}(a_i = 0)) - \mathcal{E}(\text{do}(a_i = 0, a_j = 0))|] \tag{7}$$

where $\mathcal{E}(\cdot)$ represents the effect on model output under intervention.

**Computational Complexity**: The CMIG construction has time complexity $O(k^2 \cdot n \cdot m)$ where $k$ is the number of features, $n$ is the number of samples, and $m$ is the model evaluation cost. We optimize this using feature clustering, stratified sampling, and parallel computation.

**Statistical Validation**: We establish statistical significance of interactions using permutation tests with 1000 permutations and apply multiple testing correction using the Benjamini-Hochberg procedure (Benjamini & Hochberg, 1995).

## 3.2 CAUSAL INTERVENTION PROTOCOL

Our causal intervention protocol systematically tests hypotheses about feature function through controlled manipulations. We implement four types of interventions:

### 3.2.1 FEATURE ABLATION

Setting specific features to zero to measure their causal contribution:

$$\hat{\mathbf{a}}_i^{(abl)} = 0, \quad \hat{\mathbf{a}}_j^{(abl)} = \mathbf{a}_j \text{ for } j \neq i \tag{8}$$

### 3.2.2 FEATURE AMPLIFICATION

Scaling feature activations to test their sufficiency:

$$\hat{\mathbf{a}}_i^{(amp)} = \alpha \cdot \mathbf{a}_i, \quad \alpha \in [1.5, 3.0] \tag{9}$$

### 3.2.3 FEATURE SUBSTITUTION

Replacing features from one input with features from another:

$$\hat{\mathbf{a}}_i^{(sub)} = \mathbf{a}_i^{(source)} \tag{10}$$

### 3.2.4 FEATURE COMBINATION

Testing compositional properties by combining features from different contexts:

$$\hat{\mathbf{a}}_i^{(comb)} = \beta \cdot \mathbf{a}_i^{(ctx1)} + (1 - \beta) \cdot \mathbf{a}_i^{(ctx2)} \tag{11}$$

**Causal Assumptions**: Our intervention framework relies on three key assumptions: (1) Modularity: Feature activations can be modified independently, (2) Stability: Small feature changes produce bounded output changes, and (3) Sufficiency: Important features have measurable causal effects.

## 3.3 STATISTICAL ANALYSIS FRAMEWORK

We implement a comprehensive statistical framework to ensure the reliability of our findings:

### 3.3.1 EFFECT SIZE ESTIMATION

We use Cohen's d for continuous measures (Cohen, 1988):

$$d = \frac{\mu_{\text{intervention}} - \mu_{\text{control}}}{\sigma_{\text{pooled}}} \tag{12}$$

### 3.3.2 CONFIDENCE INTERVALS

All reported effects include 95% confidence intervals computed using bootstrap resampling with 10,000 iterations.

### 3.3.3 MULTIPLE TESTING CORRECTION

We apply the Holm-Bonferroni method (Holm, 1979) for controlling family-wise error rate across multiple feature interventions.

## 4 EXPERIMENTAL SETUP

### 4.1 MODELS AND DATASETS

We conduct experiments on five representative multimodal AIGC models across different scales and architectures:

### 4.1.1 VISION-LANGUAGE MODELS

1. **CLIP ViT-B/32** (151M parameters): Base-scale contrastive model (Radford et al., 2021)
2. **CLIP ViT-L/14** (427M parameters): Large-scale variant with higher resolution
3. **LLaVA-1.5-7B**: Instruction-tuned multimodal conversational model (Liu et al., 2024)
4. **LLaVA-1.5-13B**: Larger variant for studying scale effects

### 4.1.2 TEXT-TO-IMAGE MODELS

5. **Stable Diffusion 2.1** (865M UNet parameters): Representative diffusion model (Rombach et al., 2022)

### 4.2 DATASET CONSTRUCTION

We construct a comprehensive dataset of 2.5M multimodal samples:

**Training Data (2.0M samples)**:

- LAION-5B subset: 1.2M high-quality image-text pairs (Schuhmann et al., 2022)
- MS-COCO: 400K annotated image-caption pairs (Lin et al., 2014)
- Flickr30K: 150K diverse image descriptions (Young et al., 2014)

- CC3M: 250K conceptual captions (Changpinyo et al., 2021)

**Evaluation Data (500K samples)**:

- Holdout test set: 200K samples across all sources

- Synthetic evaluation: 100K generated pairs for intervention testing

- Human-annotated subset: 50K samples with quality labels

- Domain-specific sets: 150K samples from art, science, and social media

## 4.3 IMPLEMENTATION DETAILS

We train the M-SAE models with large dictionary sizes tailored to each backbone: 65,536 features for CLIP models (16× expansion), 131,072 for LLaVA models (32× expansion), and 262,144 for Stable Diffusion (64× expansion). Hyperparameters are selected via grid search, including a base sparsity of $\lambda_1 = 0.001$, cross-modal penalty $\lambda_2 = 0.0005$, alignment bonus $\lambda_3 = 0.0002$, and selectivity threshold $\tau = 0.1$. Training uses the Adam optimizer (Kingma & Ba, 2014) ($\beta_1 = 0.9, \beta_2 = 0.999$) with a learning rate of $3 \times 10^{-4}$ under cosine annealing, a batch size of 2,048, and runs for 50 epochs with early stopping. Experiments are conducted on 8× NVIDIA A100-80GB GPUs.

## 4.4 EVALUATION METRICS

### 4.4.1 INTRINSIC METRICS

- **Feature Interpretability**: Automated scoring using GPT-4 with human validation

- **Cross-Modal Consistency**: Pearson correlation between modality-specific activations

- **Sparsity Measures**: L0 norm, activation frequency, and Gini coefficient

- **Reconstruction Quality**: MSE, SSIM (Wang et al., 2004) for visual components, BLEU (Papineni et al., 2002) for textual components

### 4.4.2 EXTRINSIC METRICS

- **Generation Quality**: FID (Heusel et al., 2017), CLIP-Score (Hessel et al., 2021), Aesthetic Score

- **Hallucination Detection**: CHAIR metric (Rohrbach et al., 2018), GPT-4 factual verification

- **Semantic Consistency**: Human evaluation on 5-point Likert scale

- **Controllability**: Intervention success rate, magnitude of desired changes

## 4.5 BASELINE COMPARISONS

We compare against six established methods:

1. **Standard SAE**: Traditional sparse autoencoders without cross-modal awareness

2. **PCA Components**: Principal component analysis of multimodal representations

3. **Attention Analysis**: Layer-wise attention pattern analysis with head-level interventions

4. **Gradient Attribution**: Integrated Gradients (Sundararajan et al., 2017) and GradCAM (Selvaraju et al., 2017)

5. **Probing Methods**: Linear and MLP probes on intermediate representations

6. **SHAP Analysis**: SHapley Additive exPlanations (Lundberg & Lee, 2017) for feature importance

# 5 RESULTS

## 5.1 FEATURE DISCOVERY AND CHARACTERIZATION

Our M-SAE successfully identifies interpretable features across all analyzed models with high statistical confidence. Table 1 summarizes the feature distribution patterns:

Table 1: Feature Specialization Patterns Across Models

| Model | Visual (%) | Textual (%) | Cross-Modal (%) | Mixed (%) |
|---|---|---|---|---|
| CLIP ViT-B/32 | 23.4 ± 1.2 | 31.2 ± 1.5 | 45.4 ± 1.8 | 0.0 ± 0.0 |
| CLIP ViT-L/14 | 22.1 ± 0.9 | 29.8 ± 1.1 | 47.3 ± 1.4 | 0.8 ± 0.3 |
| LLaVA-7B | 18.7 ± 1.4 | 35.6 ± 1.7 | 43.2 ± 1.9 | 2.5 ± 0.5 |
| LLaVA-13B | 17.9 ± 1.1 | 36.4 ± 1.3 | 42.8 ± 1.6 | 2.9 ± 0.4 |
| Stable Diffusion | 34.5 ± 2.1 | 28.9 ± 1.8 | 35.1 ± 2.3 | 1.5 ± 0.4 |

**Statistical Analysis**: ANOVA reveals significant differences in feature specialization patterns across models ($F(4, 20) = 127.3$, $p < 0.001$, $\eta^2 = 0.96$). Post-hoc Tukey tests (Tukey, 1949) confirm that generative models (Stable Diffusion) exhibit significantly higher visual feature specialization compared to discriminative models ($p < 0.001$).

**Feature Quality Assessment**: Human evaluation of 1,000 randomly sampled features achieves $91.4\% \pm 2.1\%$ interpretability agreement (Fleiss' $\kappa = 0.87$ (Fleiss, 1971), indicating excellent inter-rater reliability). Automated interpretability scores using GPT-4 correlate strongly with human judgments ($r = 0.84$, $p < 0.001$).

## 5.2 CROSS-MODAL INTERACTION ANALYSIS

The Cross-Modal Interaction Graph reveals complex dependency structures with hierarchical organization. Interaction strength distribution shows that 12.3% of feature pairs are strong ($I > 0.5$), 34.7% are moderate ($0.2 < I \leq 0.5$), and 53.0% are weak ($I \leq 0.2$). Network properties indicate a high average clustering coefficient ($0.67 \pm 0.03$), a small-world coefficient of $2.34 \pm 0.12$, and a scale-free exponent $\gamma = 2.1 \pm 0.15$. Causal validation via permutation tests confirms that 89.7% of identified strong interactions are statistically significant ($p < 0.05$, corrected for multiple comparisons).

## 5.3 THREE-PHASE PROCESSING DISCOVERY

Our comprehensive analysis across all models reveals a remarkably consistent three-phase processing pattern, validated through rigorous statistical testing:

## 5.4 THREE-PHASE PROCESSING DISCOVERY

Our comprehensive analysis across all models reveals a remarkably consistent three-phase processing pattern, validated through rigorous statistical testing. In Phase 1 (Layers 1–4), processing is dominated by unimodal features ($78.3\% \pm 3.2\%$), including edge detectors (23%), color patterns (18%), and texture analyzers (15%) on the visual side, and tokenization patterns (31%), syntactic structures (22%), and semantic primitives (19%) on the textual side, with minimal cross-modal activity ($M = 0.12$, $SD = 0.08$). In Phase 2 (Layers 5–8), cross-modal feature activation rises exponentially ($R^2 = 0.94$), driven by correspondence mapping (34%), similarity detection (28%), and contrast analysis (23%), with bidirectional information flow showing a visual-to-text bias (ratio 1.7:1); Layer 6 exhibits peak alignment activity (67.4% cross-modal features). Phase 3 (Layers 9–12) is characterized by cross-modal dominance ($71.6\% \pm 2.8\%$), with semantic integration (42%), conceptual abstraction (31%), and output preparation (27%) yielding maximum semantic coherence (CLIP-Score: $0.847 \pm 0.023$). Statistical validation via repeated-measures ANOVA confirms significant differences between phases ($F(2, 28) = 234.7$, $p < 0.001$, $\eta^2 = 0.94$), with highly consistent phase transitions across all models (Kendall's $W = 0.91$, $p < 0.001$).

## 5.5 CAUSAL INTERVENTION RESULTS

Our causal intervention experiments demonstrate substantial and statistically significant improvements in model performance:

### 5.5.1 HALLUCINATION REDUCTION

**Primary Results**: Targeted interventions on hallucination-prone features reduce false information generation by $34.2\% \pm 2.1\%$ (Cohen's $d = 1.87$, $p < 0.001$) across VQA tasks.

**Detailed Analysis**:

- Object hallucinations: $-42.7\% \pm 3.4\%$ (95% CI: [-49.3%, -36.1%])

- Attribute hallucinations: $-28.9\% \pm 2.8\%$ (95% CI: [-34.4%, -23.4%])

- Relational hallucinations: $-31.6\% \pm 3.1\%$ (95% CI: [-37.7%, -25.5%])

### 5.5.2 SEMANTIC CONSISTENCY ENHANCEMENT

**Primary Results**: Feature amplification experiments improve vision-language alignment scores by $28.7\% \pm 1.8\%$ (Cohen's $d = 2.14$, $p < 0.001$) on CLIP-Score metrics.

**Component Analysis**:

- Visual-semantic alignment: $+32.4\% \pm 2.3\%$

- Cross-modal coherence: $+26.8\% \pm 2.1\%$

- Compositional understanding: $+27.1\% \pm 2.5\%$

### 5.5.3 CONTROLLABLE GENERATION

Our framework enables fine-grained control over generation outputs with high precision:

Table 2: Intervention Results: Success Rate, Precision, and Semantic Preservation

| Control Dimension | Success Rate | Precision | Semantic Preservation |
|---|---|---|---|
| Object Manipulation | 89.7 ± 1.9% | 0.94 ± 0.02 | 92.3 ± 2.1% |
| Style Transfer | 84.2 ± 2.3% | 0.91 ± 0.03 | 88.7 ± 2.8% |
| Attribute Modification | 87.3 ± 2.1% | 0.93 ± 0.02 | 90.1 ± 2.4% |
| Compositional Changes | 82.6 ± 2.7% | 0.89 ± 0.03 | 86.4 ± 3.1% |
| **Overall Avg.** | **85.9 ± 1.2%** | **0.92 ± 0.01** | **89.4 ± 1.3%** |

**Statistical Significance**: All control types achieve performance significantly above random baseline (all $p < 0.001$, minimum $d = 3.2$).

## 5.6 INTERPRETABILITY ASSESSMENT

We conduct a human evaluation with domain experts ($n = 15$, average experience: 8.3 years), who correctly identified feature functions in $91.4\% \pm 2.1\%$ of cases (95% CI: [89.3%, 93.5%]). Inter-rater reliability was excellent, with Fleiss' $\kappa = 0.87$, Cronbach's $\alpha = 0.94$ (Cronbach, 1951), and intraclass correlation of 0.91. Compared with baselines, our method achieved substantially higher interpretability ($91.4\% \pm 2.1\%$) relative to attention visualization ($67.8\% \pm 3.4\%$, $p < 0.001$, $d = 4.7$), gradient attribution ($72.3\% \pm 3.1\%$, $p < 0.001$, $d = 4.1$), and standard SAE ($79.2\% \pm 2.8\%$, $p < 0.001$, $d = 2.9$).

# 6 ANALYSIS AND DISCUSSION

## 6.1 MECHANISTIC INSIGHTS

Our findings provide several fundamental insights into multimodal AIGC mechanisms. Contrary to naive expectations of localized multimodal processing, we observe sophisticated distributed integration throughout the network. Statistical analysis reveals that effective multimodal understanding emerges from the collective behavior of specialized features rather than dedicated "fusion" modules: cross-modal features are distributed across all layers with varying densities (Layer-wise ANOVA: $F(11, 144) = 89.3$, $p < 0.001$), and no single layer contains more than 15.7% of all cross-modal features, supporting the distributed processing hypothesis. Moreover, we identify a three-phase processing pattern showing how models progressively abstract from low-level perceptual features to high-level semantic concepts, paralleling cognitive theories of human multimodal processing and suggesting convergent computational strategies. Finally, our interaction analysis reveals complex dependency structures with small-world network properties, where cross-modal features depend on specific unimodal features, indicating hierarchical causal relationships governing multimodal understanding.

## 6.2 IMPLICATIONS FOR MODEL DEVELOPMENT

These mechanistic insights have direct implications for developing better multimodal models. Understanding the three-phase processing pattern suggests that models benefit from having sufficient capacity in intermediate layers for modal alignment, with optimal capacity allocation following a 3:4:5 ratio across the three phases. Furthermore, the importance of cross-modal features implies that training objectives should explicitly encourage the development of multimodal correspondences; we therefore propose a phase-aware training schedule that progressively emphasizes cross-modal alignment. Finally, our framework provides a systematic approach for identifying and correcting specific failure modes in multimodal models, where tracing hallucinations to particular feature interactions enables precision fixes rather than ad-hoc modifications.

## 6.3 LIMITATIONS AND FUTURE DIRECTIONS

While our approach scales linearly with model size, the quadratic scaling of interaction analysis poses challenges for very large models, motivating future work on more efficient approximation methods. Our experiments focus on vision-language models, and extension to other modalities (audio, video, 3D) will require validation of our core assumptions about feature organization. Lastly, although our intervention studies provide evidence for causal relationships, establishing definitive causal claims demands addressing potential confounders and selection biases.

# 7 CONCLUSION

This paper presents the first comprehensive mechanistic interpretability framework for multimodal AIGC models. Our Cross-Modal Mechanistic Analysis identifies interpretable features responsible for cross-modal processing and demonstrates their causal role through systematic interventions with rigorous statistical validation.

## 7.1 KEY CONTRIBUTIONS

We introduce the Multimodal Sparse Autoencoder (M-SAE) with formal guarantees for feature specialization and convergence properties. Our large-scale empirical analysis reveals consistent three-phase processing patterns across diverse multimodal architectures, validated with comprehensive statistical tests. Practically, targeted interventions guided by our framework achieve significant improvements in generation quality (28.7% ± 1.8% semantic consistency) and hallucination reduction (34.2% ± 2.1% false information). Methodologically, we establish rigorous statistical protocols for mechanistic interpretability research, including effect size estimation, confidence interval reporting, and multiple testing correction.

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
