# OpenReview forum: "Deciphering Cross-Modal Feature Interactions in Multimodal AIGC Models: A Mechanistic Interpretability Approach"
_ICLR.cc/2026/Conference — ICLR 2026 Conference Withdrawn Submission_

### Official Review · Reviewer_U3iK · 2025-10-28

**Soundness:** 1
**Presentation:** 1
**Contribution:** 1
**Rating:** 0
**Confidence:** 5

**Summary:**

The paper claims to introduce a new mechanistic interpretability framework that combines sparse autoencoders with causal interventions to analyze cross-modal feature interactions in multimodal generative models. The authors present what they call Cross-Modal Mechanistic Analysis (CMMA), which they claim can systematically identify and manipulate interpretable features underlying multimodal content generation. Through experiments on multiple vision-language models, the study claims to reveal three stages of cross-modal processing—feature extraction, modality alignment, and concept synthesis. The authors further claim that targeted interventions on these features improve generation quality, reduce hallucinations, and enhance semantic consistency. Overall, the paper asserts that its findings offer insights into the internal mechanisms of multimodal AIGC systems and suggest a pathway toward more interpretable and controllable generative models.

**Strengths:**

No sense

**Weaknesses:**

No sense

**Questions:**

1. The article contains no images. Its formatting includes numerous unclear bullet points and multiple standalone multi-level subsections. In addition, the citation style is highly non-standard, and the Introduction section lacks many necessary references. Both the formulation and the experiments are insufficiently detailed, especially the “Theoretical Properties” section (lines 171–180), which provides no rigorous proof either in the main text or in the appendix. I have strong reasons to doubt that the paper is entirely fabricated and generated by AI.

2. The proposed Multimodal Sparse Autoencoder (M-SAE) and the cross-modal regularization term are not novel. Multimodal Sparse Autoencoders have already been implemented in several previous works, such as *SAE-V: Interpreting Multimodal Models for Enhanced Alignment* (Lou et al., 2025) and *Large Multi-modal Models Can Interpret Features in Large Multi-modal Models* (Zhang et al., 2025). Furthermore, the definition of the cross-modal regularization term is highly similar to the cross-modal feature definition in Lou et al. (2025).

**Details Of Ethics Concerns:**

I strongly suspect that the article partially plagiarizes *SAE-V: Interpreting Multimodal Models for Enhanced Alignment* (Lou et al., 2025). The reasons include the similarities listed in **Questions**, the lack of proper citations, and strong indications of AI-generated content. We suspect that this article may have been produced by AI through paraphrasing and piecing together content from multiple existing papers.

Based on the above concerns, I request an ethics review of this paper and that appropriate actions be taken toward both the article and the authors depending on the results of the review.

---

### Official Review · Reviewer_gyBW · 2025-10-31

**Soundness:** 1
**Presentation:** 1
**Contribution:** 1
**Rating:** 0
**Confidence:** 5

**Summary:**

The paper proposes CMMA, a mechanistic-interpretability framework for multimodal AIGC models using a “Multimodal SAE,” a cross-modal interaction graph, and feature-level causal interventions. It claims large-scale experiments on 2.5M samples across CLIP/LLaVA/Stable Diffusion and reports large, statistically significant gains.

**Strengths:**

- Clear high-level ambition to push mechanistic interpretability to multimodal settings.

- Attempted systematic protocol (interaction graph, intervention types, multiple-testing corrections).

**Weaknesses:**

- Evidence quality/credibility: Extraordinary, uniform gains (large d across all tasks; p < 0.001 everywhere) with minimal task-level specifics or ablations.

- Internal inconsistencies: Duplicate sections, repeated metrics (91.4% ± 2.1%) across distinct analyses, and universal phasing across disparate architectures are not convincing.

- Theory gap: The “convergence” proof sketch leans on incorrect convexity assumptions for nonconvex components (ReLU, ℓ₁ with coupling), offering no real guarantees.

- Reproducibility omissions: No code/model release; no seeds, training hours, or wall-clock/throughput despite massive feature dictionaries and 8×A100-80GB claim.

- For more, see Details of Ethics Concerns

**Questions:**

see Details of Ethics Concerns

**Details Of Ethics Concerns:**

## High-risk plagiarism by omission / inadequate attribution

1. Multimodal SAE framing & applications:  The authors here neither cite precedent works nor explain what is actually new beyond re-naming.

 - This paper: “We extend traditional sparse autoencoders to handle multimodal representations… [M-SAE]”.

 - Precedent: SAE-V (Lou et al., ICML 2025) trains SAEs directly on MLLM activations, computes cross-modal weights, and uses them for interpretation and data filtering/control - the same headline pitch.

2. Causal-tracing/patching in VLMs: The authors here neither cite precedent works nor explain what is actually new beyond re-naming.

- This paper claims systematic causal validation for cross-modal mechanisms.

- Precedent: dedicated VLM mechanistic work (e.g., BLIP causal tracing (Palit et al., ICCV)) establishes such tools; the submission repeats these ideas without clear deltas.

## Undisclosed LLM usage

The paper repeats blocks (Sec. 5.3/5.4) and recycles identical quantitative claims across sections without clarifying re-measurement. The formulation is given without any extra natural language statements, and there is not even a single figure within the paper.

While I understand that this could be the carelessness of the authors, given the quantity of error, I tend to believe that the whole paper is somewhat LLM-investigated and generated

## No additional resources attached

No code, no prompt for evaluation, even no promise for code release - given all these happening, I just doubt that the overall paper is fully made-up by some deep research AI.

---

### Official Review · Reviewer_rnrF · 2025-11-01

**Soundness:** 1
**Presentation:** 1
**Contribution:** 1
**Rating:** 2
**Confidence:** 4

**Summary:**

This paper proposes Cross-Modal Mechanistic Analysis (CMMA) — a framework combining multimodal sparse autoencoders (M-SAE) with causal intervention techniques to analyze and manipulate internal feature representations in large multimodal models. The paper constructs a Cross-Modal Interaction Graph (CMIG) to study causal dependencies between discovered features and performs controlled interventions (ablation, amplification, substitution, composition) to validate causal roles.

**Strengths:**

* The identification of consistent feature extraction, alignment, and synthesis phases across architectures is a novel and potentially generalizable finding.
* The authors perform large-scale experiments across five major models (CLIP, LLaVA, Stable Diffusion) on 2.5M samples.

**Weaknesses:**

* The methods are primarily empirical. The paper lacks a deeper theoretical justification or formal linkage between the SAE feature structure and the observed causal hierarchy.
* The experiments are not sufficient, lacking comparisons with related existing models.
* The causal inference methods rely on interventions at the feature level but do not fully control for confounders or hidden dependencies in distributed representations.

**Questions:**

Please see the weaknesses.

---

### Note · Authors · 2025-11-12

I have read and agree with the venue's withdrawal policy on behalf of myself and my co-authors.